# Investigation of Hydrodynamic Parameters in an Airlift Photobioreactor on CO₂ Biofixation by *Spirulina* sp.

**Zahra Zarei [1], Peyman Malekshahi [1], Antoine P. Trzcinski [2] and Mohammad Hossein Morowvat [3,4,*]**

[1]  Department of Chemical Engineering, University of Sistan & Baluchestan, Zahedan P.O. Box 98167-45845, Iran; zahrazarei02@gmail.com (Z.Z.); malekshahipayman@gmail.com (P.M.)

[2]  School of Civil Engineering and Surveying, University of Southern Queensland, West Street, Toowoomba QLD, 4350, Australia; antoine.trzcinski@usq.edu.au

[3]  Pharmaceutical Sciences Research Center, Shiraz University of Medical Sciences, Shiraz P.O. Box 71468-64685, Iran

[4]  Department of Pharmaceutical Biotechnology, School of Pharmacy, Shiraz University of Medical Sciences, Shiraz P.O. Box 71468-64685, Iran

[*]  Correspondence: mhmorowvat@sums.ac.ir; Tel./Fax: +98-71-324-267-29

**Abstract:** The rise of $CO_2$ concentration on Earth is a major environmental problem that causes global warming. To solve this issue, carbon capture and sequestration technologies are becoming more and more popular. Among them, cyanobacteria can efficiently sequestrate $CO_2$, which is an eco-friendly and cost-effective way of reducing carbon dioxide, and algal biomass can be harvested as valuable products. In this study, the hydrodynamic parameters of an airlift photobioreactor such as gas holdup, mean bubble diameter and liquid circulation velocity were measured to investigate $CO_2$ biofixation by *Spirulina* sp. The total gas holdup was found to increase linearly with the increase in the gas velocity from 0.185 to 1.936 cm/s. The mean bubble velocities in distilled water only and in the cyanobacterial culture on the first and sixth days of cultivation were 109.97, 87.98, and 65.89 cm/s, respectively. It was found that shear stress at gas velocities greater than 0.857 cm/s led to cyanobacterial death. After 7 days of batch culture, the maximum dry cell weight reached 1.62 g/L at the gas velocity of 0.524 cm/s, whereas the highest carbon dioxide removal efficiency by *Spirulina* sp. was 55.48% at a gas velocity of 0.185 cm/s, demonstrating that hydrodynamic parameters applied in this study were suitable to grow *Spirulina* sp. in the airlift photobioreactor and remove $CO_2$.

**Keywords:** airlift photobioreactor; carbon sequestration; $CO_2$ biofixation; gas holdup; liquid circulation velocity; *Spirulina* sp.

## 1. Introduction

Global warming caused by human activities is one of the significant environmental problems that has received much attention in the last two decades. One of the reasons for the increase in temperature and global warming is the presence of greenhouse gases, especially the high concentration of carbon dioxide in the Earth's atmosphere [1]. This concentration was 260–280 ppm before the industrial revolution, and it has increased significantly since then, reaching 381 ppm in 2005. In the last 100 years, the air temperature has increased between 0.18 and 0.74 °C [2]. A few researchers reported an annual increase of 0.2 °C in the Earth's temperature over the last three decades due to $CO_2$ emissions, which reached a new record (37%) compared to the late 18th century [3].

Among various methods for reducing $CO_2$ such as physical absorption, chemical fixation, soil carbon sequestration, biochar and $CO_2$-enhanced oil recovery, the biological fixation of $CO_2$ by photosynthetic cyanobacteria have gained attention as an environmentally friendly and economically competitive technology [4–7]. The process requires knowledge of cell biology and factors influencing this process such as temperature, pH, light, cyanobacteria species, cyanobacterial culture density, $CO_2$ concentration, photobioreactor hydrodynamic parameters (reactor configuration, sparging rate, mixing) [8,9].

However, $CO_2$ fixation by cyanobacteria on a large scale remains challenging due to the limitation of cyanobacterial growth at high concentrations of $CO_2$ and the need for more nutrients and light [10,11]. Therefore, selecting suitable cyanobacteria species to thrive at high $CO_2$ concentrations is considered an essential step in the biological removal of $CO_2$ [10,12]. *Spirulina* sp. is an ideal candidate due to its resistance to harsh environmental conditions, high $CO_2$ fixation ability, and growth rate [13,14]. For instance, the removal efficiency of $CO_2$ fixation by *Spirulina platensis* at 5% $CO_2$ in the feed gas was found to be 81% in a semi-continuous process [15]. Also, using 6% and 12% $CO_2$, the maximum $CO_2$ biofixation by *Spirulina* sp. was 53.3% and 45.6%, respectively [16]. Other researchers observed that the maximum $CO_2$ removal efficiency by *S. platensis* at 4% and 6% $CO_2$ was 70% and 65%, respectively, in a tubular photobioreactor [17,18]. Soletto et al. [19] reported that the maximum carbon dioxide removal efficiency of *S. platensis* at 80 and 125 mol photons $m^{-2}$ $s^{-1}$ reached 50% and 69%, respectively, in a bench-scale helical photobioreactor; however, at 166 mol photons $m^{-2}$ $s^{-1}$, the $CO_2$ biofixation was higher than 90%.

Typical photobioreactors (PBRs) such as vertical column reactors (bubble column and airlift reactors), flat plate, and tubular reactors have been used to achieve high productivity, uniform mixing, high area-to-volume ratio, high light utilization efficiency, and adequate temperature and light control which are desirable characteristics [20,21]. Airlift reactors (ALRs) are ideal for $CO_2$ biofixation due to good mixing, high mass transfer rate, low shear stress. They also require lower energy input and can be used in a wide range of inlet gas flowrates and higher viscosity fluids [22–24]. Like in bubble columns, agitation is achieved through pumping air which provides low shear stress and high mass transfer [25,26]. ALRs with an internal loop consist of 4 main parts:

(i) the riser (draft tube) is the most important part of the reactor where most of the mixing and mass transfer occurs. The gas distributor is located at the bottom of this section and a directional multiphase flow is created upwards in this area.

(ii) the downcomer works in parallel with the riser and is the external tube. The density in this part is higher than the riser, which causes the fluid to move.

(iii) the sparger (or gas distributor) is located at the bottom of the reactor where the riser and the downcomer are connected.

(iv) the gas separator connecting the two sides of the reactor at the top and the gas separation operation is performed there [24,25]. In airlift reactors, the gas holdup, which depends on reactor height and the gas-liquid separation area in the reactor headspace, is one of the key parameters in determining the liquid circulation velocity, gas residence time, and overall mass transfer coefficient [24,27,28].

Furthermore, gas holdup is affected by the gas velocity, which is known as gas flow rate. As the gas flow rate increases, the gas holdup in the riser and downcomer increases, and as a result, the liquid circulation and bubble velocities increase. Larger bubbles move faster and descend to the downcomer, which results in better light absorption by cyanobacteria [29,30]. However, as the liquid velocity increases at higher gas flow rates, cyanobacteria do not remain in the downcomer long enough to utilize light efficiently [31] and may suffer from shear stress. Liquid velocity affects many hydrodynamic and mass transfer parameters, including average bubble residence time, bubble size, overall mass transfer rate, and mixing time. In addition, it influences turbulence, mass transfer coefficient, and shear stress [25,29]. As a result, there is a lack of data and studies on the effect of hydrodynamic parameters of the ALR (gas holdup, bubble diameter, and gas and liquid velocities) on the growth of cyanobacteria and $CO_2$ biofixation. Therefore, the present study aims at evaluating the $CO_2$ biofixation by *Spirulina* sp. through the optimization of gas flowrate, gas holdup, and liquid velocities in an airlift photobioreactor.

## 2. Materials and Methods

### 2.1. Algal Strain and Cultivation Conditions

*Spirulina* sp. strain was sourced from Science and Technology Park, Bushehr, Iran, and it was cultivated in BG-11 culture medium [32], which is described in Table 1. The stock

culture was maintained in a 500 mL Erlenmeyer flask containing 250 mL of the medium at 27 °C under 60 μmol $m^{-2}$ $s^{-1}$ of light intensity and a 12/12 h day/night cycle [15].

**Table 1.** Composition of BG-11 culture medium.

| Stock Solution | Composition | Stock Solution g $L^{-1}$ |
|---|---|---|
| Nutrient Solution of BG-11 culture medium | $NaNO_3$ | 1.5 g |
| | $K_2HPO_4$ | 0.04 g |
| | $MgSO_4 \cdot 7H_2O$ | 0.075g |
| | $CaCl_2 \cdot 2H_2O$ | 0.036 g |
| | citric acid | 6.0 mg |
| | ferric ammonium citrate | 6.0 mg |
| | $Na_2EDTA$ | 1.0 mg |
| | $Na_2CO_3$ | 0.02 g |
| Trace metals mix A5 (1 mL for 1 L BG-11) | $H_3BO_3$ | 2.86 g |
| | $MnCl_2 \cdot 4H_2O$ | 1.81 g |
| | $ZnSO_4 \cdot 7H_2O$ | 0.222 g |
| | $Na_2MoO_4 \cdot 2H_2O$ | 0.39 g |
| | $CuSO_4 \cdot 5H_2O$ | 0.079 g |
| | $Co(NO_3)_2 \cdot 6H_2O$ | 49.4 mg |

### 2.2. Airlift Photobioreactor Configuration

The airlift photobioreactor used in this experiment, as shown in Figure 1, was made of two vertical Plexiglas loops with a stainless-steel circular distributor installed at the bottom of the reactor. The distributor consisted of eighteen 1-mm holes placed at regular intervals.

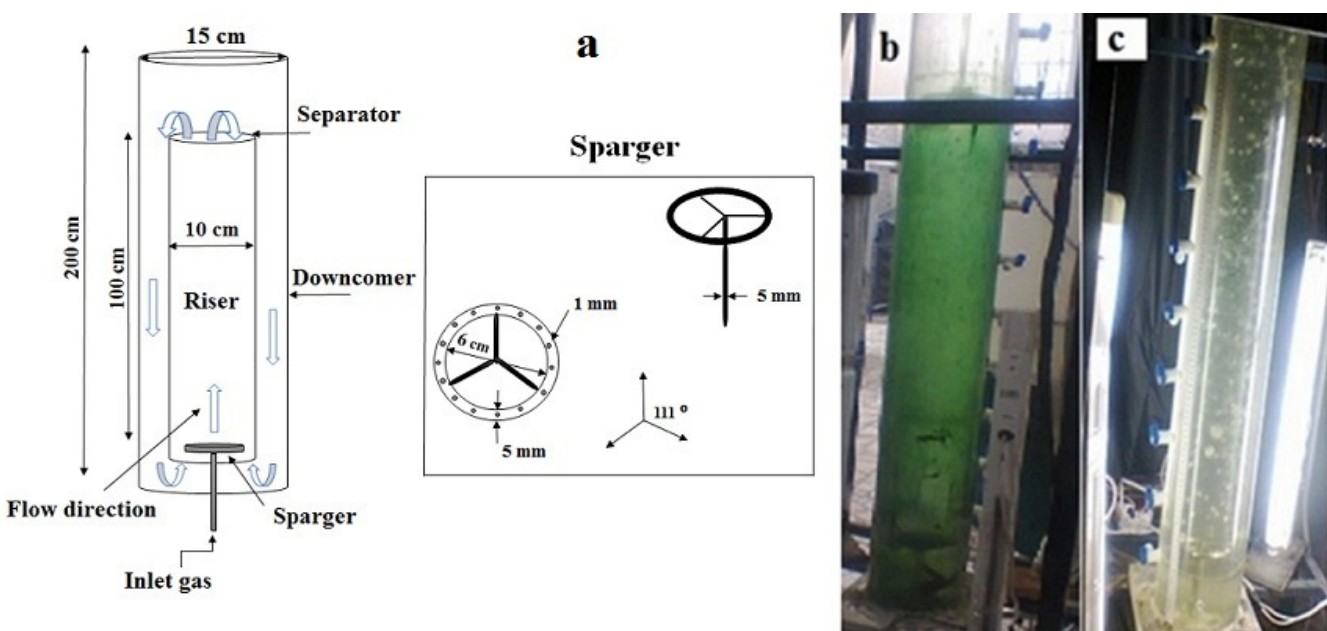

**Figure 1.** (**a**) Schematic diagram of the experimental setup; Images of (**b**) microalgae growth and (**c**) distilled water in the airlift photobioreactor.

The total volume and working volume of the reactor were 20 L and 16 L, respectively. To control the temperature at 30 °C, a 100-watt heater (Sobo HC-100 model) was fitted inside the reactor. Also, two gas rotameters were used to measure the air and carbon

dioxide flow rates. Illumination was provided using four fluorescent lamps, and light intensity was measured with a lux meter (TES-1339R, TES Taiwan, Taipei, Taiwan).

### 2.3. Hydrodynamic Tests in Distilled Water

Initially, the hydrodynamic parameters of ALR were investigated by performing gas holdup tests in distilled water only by varying the inlet gas velocity and measuring the average bubble size. For this purpose, experiments, which were repeated three times, conducted at six different gas velocities of (0.185, 0.524, 0.857, 1.176, 1.593, and 1.936 cm/s) (Table 2), to find out the range of the bubbly flow (homogenous regime) for optimum cyanobacterial growth.

**Table 2.** Hydrodynamic parameters tested in the ALR at different gas velocities.

| Gas velocity (cm/s) | 0.185 | 0.524 | 0.857 | 1.176 | 1.593 | 1.936 |
|---|---|---|---|---|---|---|
| Gas holdup (cm/cm) | | | | | | |
| Mean bubble diameter (mm) | | | | | | |
| Bubble velocity (cm/s) | | | | | | |
| Liquid circulation velocity in riser (cm/s) | | | | | | |
| Liquid circulation velocity in downcomer (cm/s) | | | | | | |

### 2.4. Inoculation and Operation of the Photobioreactor

*Spirulina* sp. growth and $CO_2$ removal efficiency were studied after transferring 1.6 L of an acclimated culture of *Spirulina* sp. into the ALR and filling it with 14.4 L of culture medium. A mixture of 95% air and 5% (*v/v*) $CO_2$ [15,33] was sparged at the bottom of the reactor. Two separate experiments were carried out for 7 days at gas velocities of 0.185 and 0.524 cm/s corresponding to flow rates of 0.87 and 2.47 L/min or 0.054 and 0.154 volume of air per medium volume per minute (vvm), respectively. The initial concentration of cyanobacteria was about 0.5 g $L^{-1}$. The light intensity was 150 µmol photons $m^{-2}$ $s^{-1}$ with a 12:12 light-dark (LD) cycle, and the temperature was $30 \pm 1\ °C$ [15,34]. The pH of the culture medium was adjusted to $9.5 \pm 0.02$ using 6 M NaOH [19,35]. The pH was measured using a pH meter (Metrohm 827 pH Lab meter). BG-11 medium was added weekly from the top of the reactor equipped with a small tank with a one-way valve to compensate for water evaporation from the photobioreactor.

### 2.5. Gas Holdup and Bubble Size Measurements

The gas holdup is described as a fraction of a gas-filled reactor volume and calculated from the following Equation (1) [25]:

$$\varepsilon = \frac{H_2 - H_1}{H_2} \qquad (1)$$

$H_2$ and $H_1$ are liquid heights after aeration and initial in the reactor, respectively.

This parameter affects the residence time of the gas phase in the liquid phase, and it influences the gas-liquid interface through bubble size. In this study, the total gas holdup test was performed using distilled water and air first, then with the cyanobacterial culture aerated with air and carbon dioxide. Since bubbles with different sizes exist in the dispersed phase, the Sauter mean bubble diameter ($d_{32}$), which is a sphere diameter with the same volume to surface ratio as the whole bubble size, was used in calculations and is determined by Equation (2) [25,36,37]:

$$d_{32} = \frac{\sum_{i=1}^{N} N_i d_i^3}{\sum_{i=1}^{N} N_i d_i^2} \qquad (2)$$

In which $N_i$ is the occurrence of the bubble with diameter $d_i$. Also, the bubble velocity was calculated using Equation (3) [25]:

$$U_b = \frac{U_g}{\varepsilon} \tag{3}$$

where $U_g$ is the gas velocity.

In this experiment, the average bubble size was analyzed using Image J software.

### 2.6. Gas Velocity

The gas velocity directly impacts the gas holdup, which is more apparent in bubbly regimes than in turbulent flow. In the airlift reactor, the gas velocity is equal to the ratio of the gas inlet volume to the riser cross-section as shown in Equation (4) [36]:

$$U_g = \frac{G_g}{A_r} \tag{4}$$

where $G_g$ and $A_r$ are the inlet flow rate to the airlift reactor and riser cross-section, respectively.

### 2.7. Liquid Circulation Velocity

Due to the difference in fluid density between riser and downcomer in reactors, the fluid reaches a certain velocity [38]. The mean circulation velocity is defined by Equation (5) [39]:

$$U_{\text{circ}} = \frac{X_{\text{circ}}}{t_{\text{circ}}} \tag{5}$$

where $x_c$ and $t_c$ are the length of the circulation path and the mean circulation time, respectively, however, liquid velocity values in riser ($U_r$) and downcomer ($U_d$) are more useful than mean circulation velocity ($U_{\text{circ}}$). However, the liquid velocity in the riser is calculated with Equation (5) and the following correlation (Equation (6)) was used to determine the liquid velocity in the downcomer in which $A_d$ and $A_r$ are downcomer and riser cross-sections, respectively [39].

$$U_d \times A_d = U_r \times A_r \tag{6}$$

### 2.8. Calculation of $CO_2$ Removal Efficiency

To estimate the $CO_2$ removal efficiency, Equation (7) was used;

$$CO_2 \text{ removal efficiency } (\%) = \left( 1 - \frac{CO_2 \text{ output}}{CO_2 \text{ input}} \right) \times 100 \tag{7}$$

where $CO_2$ input is the carbon dioxide concentration in the gas injected into the bioreactor, while $CO_2$ output is the concentration in ppm of carbon dioxide exiting the reactor. To measure $CO_2$ concentration in the headspace of the reactor, a $CO_2$ meter (Testo, Lenzkirch, Germany, model 535) was used.

### 2.9. Biomass Concentration

Each day, a 20 mL sample from the ALR was withdrawn to measure the growth medium optical density at 680 nm ($OD_{680}$) [15,40] using a spectrophotometer (DR 5000, Hach, Loveland, HQ, USA).

To obtain the dry cell weight, the sample was centrifuged at 2000 rpm for 15 min [41] using a centrifuge (Sigma 3-30KS, Sigma, Burlington, MA, USA). The residue was then placed in an oven for 72 h to remove moisture completely. Finally, the total dry weight of *Spirulina* sp. was measured.

Specific growth rate ($\mu$, day$^{-1}$) was calculated according to Equation (8):

$$\mu = \frac{\ln(X_2) - \ln(X_1)}{\Delta t} \tag{8}$$

where $X_2$ and $X_1$ represent biomass concentration (g L$^{-1}$) during culture days $t_1$ and $t_2$. To estimate the biomass productivity ($p$, g L$^{-1}$ day$^{-1}$), Equation (9) was used.

$$p = \frac{X_2 - X_1}{t_2 - t_1} \tag{9}$$

### 3. Results and Discussion

*3.1. Gas Holdup and Bubble Behavior in Distilled Water*

As shown in Figure 2, it was observed that an increase in the gas velocity from 0.185 to 1.936 cm/s resulted in a relatively small increase in gas holdup compared to other empirical correlations. To predict gas holdup in the ALR in the rage of gas velocity studied, theoretical correlations [42–46] which are presented in Table 3, were used. As in this study Equation (1) was used, factors such as gas and liquid densities ($\rho_g$ and $\rho_l$), the surface tension of the liquid ($\sigma$) gas and liquid viscosities ($\mu_g$ and $\mu_l$) and gravitational constant (g) were ignored, and as a result, there is a difference between measured gas holdup and predicted holdup using empirical correlations [42–45]. However, the experimental data obtained in this study is consistent with Hikita et al.'s correlation [46]. They confirmed that total gas holdup increases linearly with the inlet gas velocity in the range of velocity investigated. The reason for this was the change in the shape and nature of gas bubbles at a higher velocity of the inlet gas [30,47,48]. As gas velocity increased, bubble size also increased (larger bubbles were released from the sparger at the bottom), but as these large bubbles rose in the liquid column, they quickly broke down into small bubbles; as a result, the number of smaller bubbles increased and the mean bubble diameter decreased, which is presented in Table 4; also the accumulation of large bubbles in the center and small ones along the walls of the reactor was observed. Chisti reported that in the total gas holdup, the percentage of small bubbles is greater than large bubbles [49].

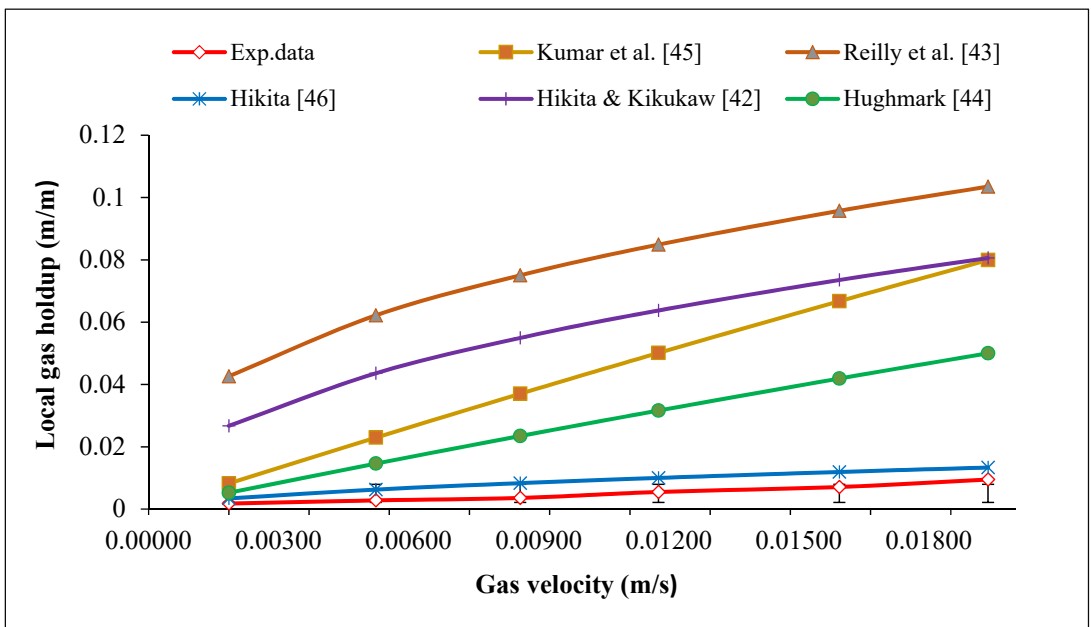

**Figure 2.** Comparison of gas holdup experimental data with empirical correlations [42–46].

**Table 3.** Theoretical correlations used in this study.

| Hikita and Kikukaw [42] | $\varepsilon = 0.505 u_g^{0.47} \left( \frac{0.072}{\sigma} \right)^{0.67} \left( \frac{0.001}{\mu_l} \right)^{0.05}$ |
|---|---|
| Reilly et al. [43] | $\varepsilon = 0.009 + 296 u_g^{0.44} \rho_l^{-0.98} \sigma_l^{-0.16} \rho_g^{0.19}$ |
| Hughmark [44] | $\varepsilon = \frac{1}{2 + (0.35/u_g)(\rho_l \sigma/72)^{0.33}}$ |
| Kumar et al. [45] | $\varepsilon = 0.728u - 0.485u^2 + 0.0975u^3$ <br> $u = u_g \left[ \frac{\rho_l^2}{\sigma} (\rho_l - \rho_g) g \right]^{1/4}$ |
| Hikita [46] | $\varepsilon = 0.672 (u_g \mu_l)^{0.578} \left( \frac{\mu_l^4 g}{\sigma^3 \rho_l} \right)^{-0.131} \left( \frac{\rho_g}{\rho_l} \right)^{0.062} \left( \frac{\mu_g}{\mu_l} \right)^{0.107}$ |

**Table 4.** Mean bubble diameter at different gas velocities in the airlift photobioreactor.

| Gas velocity (cm/s) | 0.185 | 0.524 | 0.857 | 1.176 | 1.593 | 1.936 |
|---|---|---|---|---|---|---|
| Flow rate (L/min) | 0.87 | 2.47 | 4.03 | 5.54 | 7.50 | 9.12 |
| vvm | 0.054 | 0.154 | 0.25 | 0.343 | 0.468 | 0.568 |
| Mean bubble diameter in distilled water (mm) | $12.8 \pm 0.64$ | $10.4 \pm 0.52$ | $9.9 \pm 0.49$ | $9.2 \pm 0.46$ | $5.9 \pm 0.29$ | $4.6 \pm 0.23$ |
| Mean bubble diameter in microalgal culture (mm) | 11.1 | 8.6 | 7.8 | 7.3 | 4.2 | 3.8 |

In Figure 3, the results of total gas holdup in cyanobacterial solution were compared with that obtained using distilled water (Figure 2). The mean bubble velocities in distilled water and cyanobacterial medium on the first and sixth days of cultivation were 109.97, 87.98, and 65.89 cm/s, respectively. Results indicated that the bubble velocity in both distilled water and cyanobacterial culture increased with increasing gas velocity, and bubbles were widely distributed. However, as the size of the bubbles directly influences the bubble velocity, higher bubble velocity was observed in distilled water compared to cyanobacterial culture due to a lower viscosity. This observation is in agreement with previous studies [50,51].

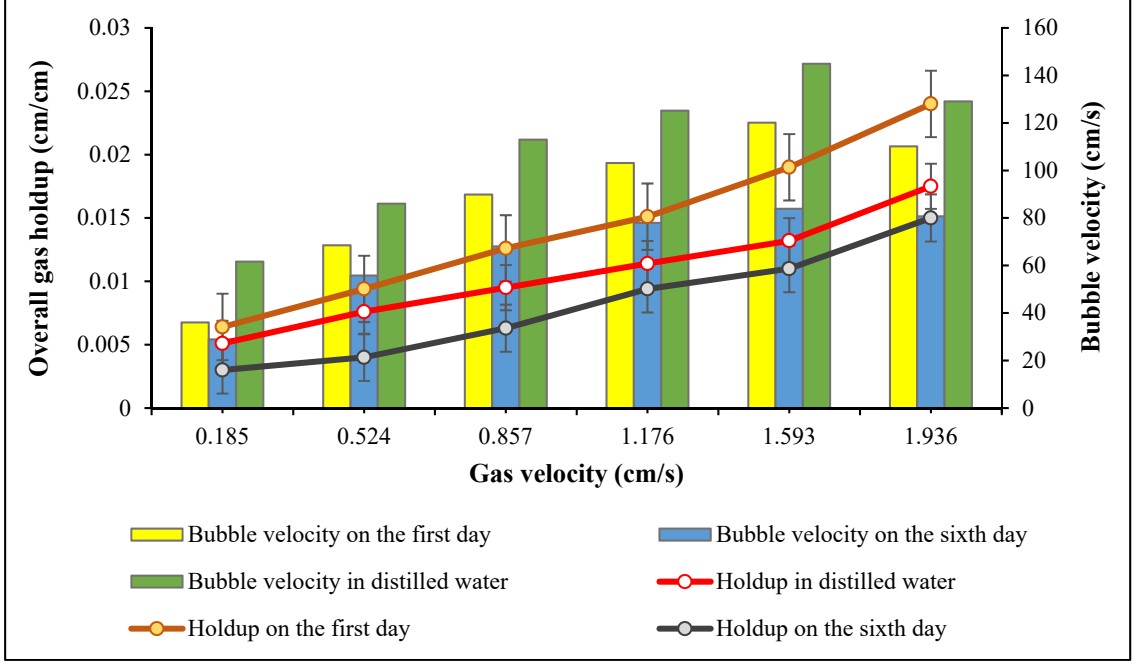

**Figure 3.** Comparison of gas holdup and bubble velocity at different gas velocities in distilled water, and on the first and sixth days of cyanobacterial growth in the reactor.

### 3.2. Gas Holdup in Cyanobacterial Culture

In the experiment using cyanobacterial culture, the total gas holdup was measured on two different days. The experimental results aligned with Equation (3), which clearly shows that the total holdup decreases when the bubble velocity increases at a constant gas velocity [25]. Also, the gas holdup on the first day was higher than in distilled water. According to Chisti [25], at the beginning of the cell growth, smaller bubbles are formed, which increase gas holdup. Also, the ions dissolved in the cyanobacterial culture medium prevented the bubbles from coalescing and forming larger bubbles [25]. As growth took place in the airlift reactor, the holdup decreased. On the sixth day, when *Spirulina* sp. was at its highest density, there was less turbulence in the ALR, resulting in a lower gas holdup, which is consistent with other works [22,36]. According to some researchers [52], gas holdup on the first day of microalgal growth was 48.7% higher than the holdup in distilled water, and with the increase in the liquid viscosity, holdup declined. Another study [53] investigated the impact of high liquid viscosity on gas holdup in a cylindrical split airlift photobioreactor. They observed a decline in the holdup with an increase in viscosity.

### 3.3. Liquid Circulation Velocity

The movement of gas bubbles pushes the liquid upwards in the riser and falls in the downcomer. As shown in Figure 4, the increase in the gas velocity resulted in an increase in the liquid velocity of the riser and downcomer, but these changes were faster at low gas velocities. This observation is in close agreement with other studies [39,54].

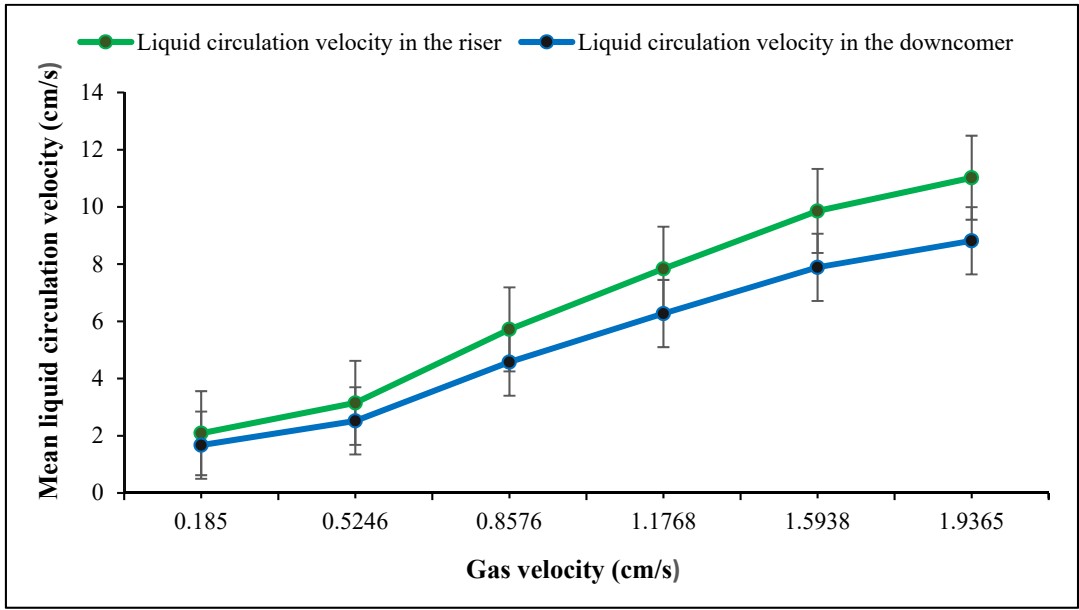

**Figure 4.** Comparison of liquid circulation velocity in the riser and downcomer at 6 different gas velocities.

### 3.4. Cyanobacterial Growth

It was observed that increasing the inlet gas velocity to 1.936 cm/s was detrimental as it resulted in a major decrease in cyanobacterial growth. The decline began at a flowrate of 0.857 cm/s when the flow became more turbulent. With the increase in the aeration rate, the mean bubble diameter gradually decreased from 11.1 to 3.8 mm (Table 4). As higher velocities caused more turbulence, the average bubble diameter was smaller due to micro-eddies, which caused small bubbles, and damaged cells which is in agreement with previous work [30]. It was reported that the mean bubble size is important for better cyanobacterial growth because with an increase in the bubble size, algal growth can also decline [25]. Moreover, Sadeghizadeh et al. studied the effect of input gas velocity on *Chlorella vulgaris* growth in an ALR, they found that with an increase in the gas velocity

over 0.74 cm/s biomass concentration declined [55]. As shown in Figures 5–7, the dry cell weight, specific growth rate and biomass productivity of *Spirulina* sp. were analyzed daily during separate experiments carried out at two different gas velocities (0.185, 0.524 cm/s). As shown in Figure 5, due to some settling in the reactor after the inoculation, the dry cell weight decreased on day 1 compared to day 0, also some cells got stuck on the reactor walls. However, the biomass concentration increased steadily during the culture period and reached a maximum dry cell weight of 1.62 g/L at the gas velocity of 0.524 cm/s, which is presented in Table 5. Increasing the aeration rate resulted in an increase in the gas holdup and fluid circulation rate, which improved the mass transfer and cyanobacterial growth in the bioreactor, which is in line with a previous study [39]. At 0.524 cm/s, the sedimentation rate reduced, and more cyanobacteria were suspended in the culture medium. As a result, cells had better exposure to light, which caused better light absorption, photosynthesis, and growth rate. Thus, 0.524 cm/s was considered a more suitable gas velocity for *Spirulina* sp. without damaging cells. It was also observed that there was a statistically significant difference between group means as determined by one-way ANOVA ($F = 5.28414$, $p = 0.04029$). In a similar study [30], which was carried out in a 50 L airlift photobioreactor using 5% input $CO_2$ and at a flow rate of 12.0 L/min, the maximum dry cell of *Chlorella protothecoides* was reported to be 0.24 g/L after 300 h. They found that when the aeration rate was 8.0 L/min, the final biomass concentration reached 0.32 g/L after 500 h (Table 5) highlighting the effect of the aeration rate on cell shear stress. Another work investigated the impact of increasing aeration rate from 0.145 vvm to 0.29 vvm with a mixture of 0.038% $CO_2$ and air on the growth rate of *C. vulgaris* in an ALR with a working volume of 80 L. The maximum biomass concentration of 1.42 g/L was achieved at an aeration rate of 0.29 vvm after 14 days (Table 5) [56]. Also, it was reported that the maximum biomass concentration of *Chlorella* sp. with a constant $CO_2$ concentration of 1.75% reached 2.8 g/L in a bubble column reactor after 13 days. They found that with an increase in the flow rate from 50 to 70 mL/min, the maximum biomass concentration of *Chlorella* sp. increased from 2.5 to 2.8 g/L (Table 5) [57].

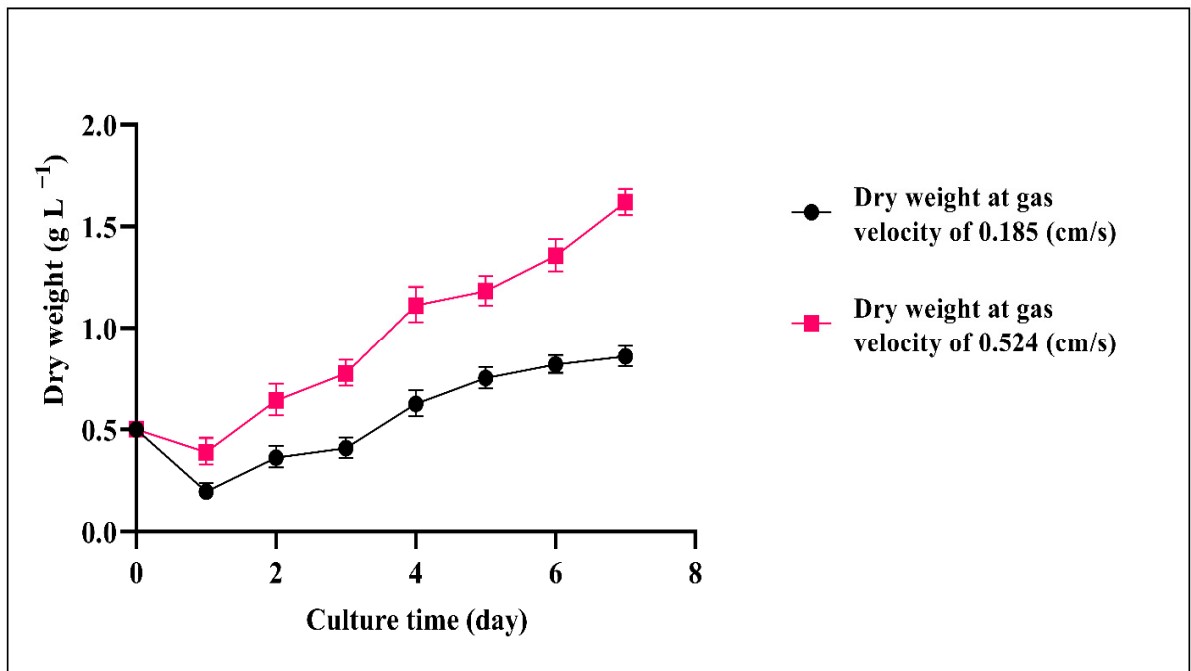

**Figure 5.** Dry weight of *Spirulina* sp. at two gas velocities (0.185, 0.524 cm/s), temperature 30 ± 1 °C and pH = 9.5 ± 0.02.

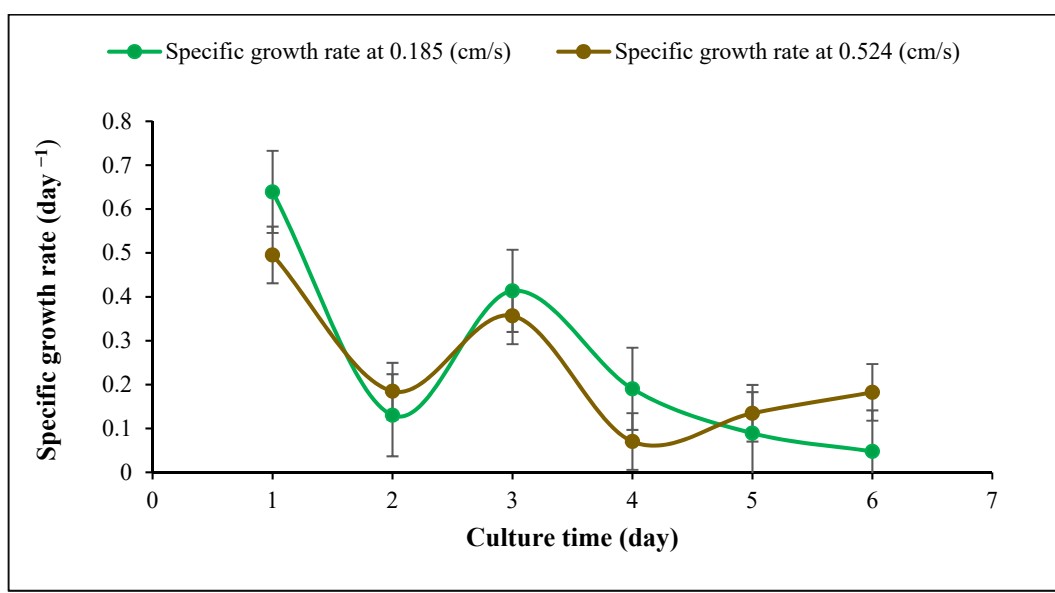

**Figure 6.** Specific growth rate of *Spirulina* sp. at two gas velocities (0.185, 0.524 cm/s), temperature 30 ± 1 °C and pH = 9.5 ± 0.02.

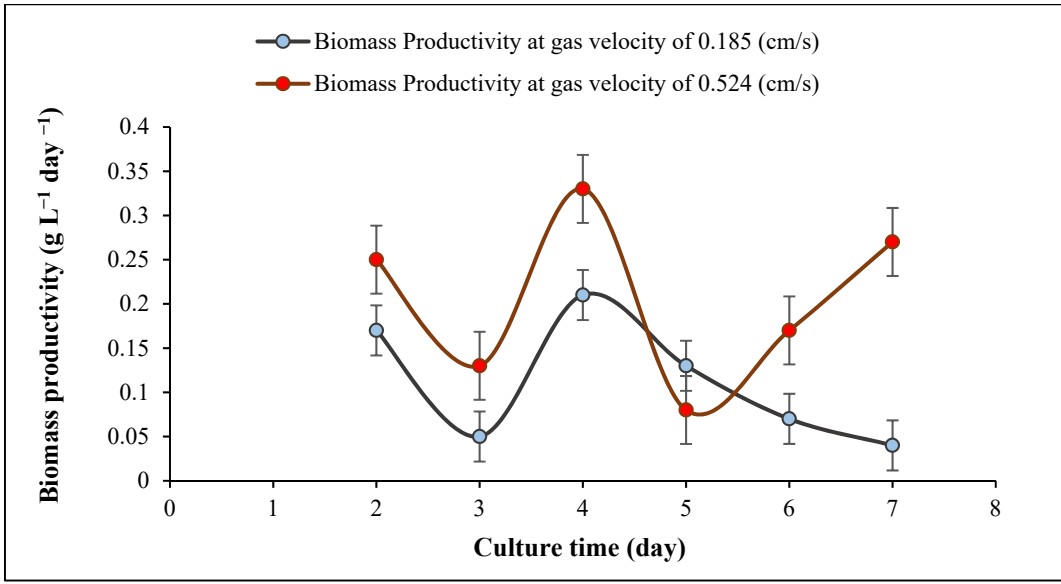

**Figure 7.** Biomass productivity of *Spirulina* sp. at two gas velocities (0.185, 0.524 cm/s), temperature 30 ± 1 °C and pH = 9.5 ± 0.02.

### 3.5. $CO_2$ Removal

Figure 8 shows the carbon dioxide removal efficiency at each gas velocity during 7 days. The results demonstrated that during the first four days, the $CO_2$ removal efficiency for both velocities increased gradually; however, during the last three days, there was a sharp rise at the gas velocity of 0.185 cm/s. Although the gas velocity of 0.524 cm/s provided better gas-liquid mixing, recirculation, and cyanobacterial growth, the highest carbon dioxide removal efficiency (55.5%) (Table 5) was achieved at the gas velocity of 0.185 cm/s and a dry cell weight of 0.86 g/L. The results illustrated that there was a statistically significant difference between group means as determined by one-way ANOVA (F = 6.15675, $p$ = 0.02889). This result indicated that higher $CO_2$ removal efficiency was achieved at the lower aeration rate for several reasons. Firstly, $CO_2$ capture efficiency can decrease because bubbles interconnect at higher gas velocity. Bubble surface area per unit volume of gas decreased,

and larger bubbles rose faster than smaller ones [22,58]; as a result, the $CO_2$ uptake from gas bubbles reduced, which is in agreement with previous research [49]. Secondly, when the inlet gas velocity increased, $CO_2$ fixation in the high-density culture (at higher viscosity) declined due to a decrease in gas retention time and $CO_2$ consumption [59]. In a study carried out at four flow rates (0.145, 0.195, 0.24, and 0.29 vvm) using *C. vulgaris* in an airlift reactor, it was found that $CO_2$ removal efficiency decreased from 28% at 0.145 vvm to 2% at 0.29 vvm (Table 5) due to a shorter residence time at higher gas velocity [56]. Also, it was reported that at two inlet gas velocities (0.74 and 1.32 cm/s) with 2% $CO_2$ in an ALR, the maximum $CO_2$ removal efficiency of *C. vulgaris* was 80% at 0.74 cm/s (Table 5) [55]. A similar work carried out in a bubble column reactor by sparging 4% $CO_2$ indicated that with the increase in superficial gas velocity from 0.10 to 0.50 cm/s, the $CO_2$ removal efficiency of *C. vulgaris* declined from 14.6% to 3.8% (Table 5) [60]. Li et al. [61] also studied the impact of increasing gas aeration rate on $CO_2$ removal efficiency of *Scenedesmus obliquus* WUST4 enriched with 12% $CO_2$ in an airlift photobioreactor and found that the maximum $CO_2$ removal efficiency was 67%, which was achieved at a flow rate of 0.1 vvm, but it decreased to 20% when increasing the aeration rate to 0.5 vvm (Table 5). However, some authors [58] claimed that increasing cell density, which led to an increase in the culture viscosity, decreased $CO_2$ concentration in the headspace of the reactor due to an increase in the gas retention time and carbon dioxide consumption. Nevertheless, with the increase in biomass concentration, the amount of light absorbed by cells diminished due to the higher viscosity of the culture medium. It means that the available light limits the rate of photosynthesis, which in turn reduces the $CO_2$ consumed by microalgae [62]. Our results are consistent with the previous work carried out by Pourjamshidian et al. [57], where the authors observed that the highest $CO_2$ removal efficiency of *Chlorella* sp. was 95.45% at a flow rate of 50 mL/min, and this declined to 88.63% at an aeration rate of 70 mL/min due to a limited algal photosynthetic capacity (Table 5).

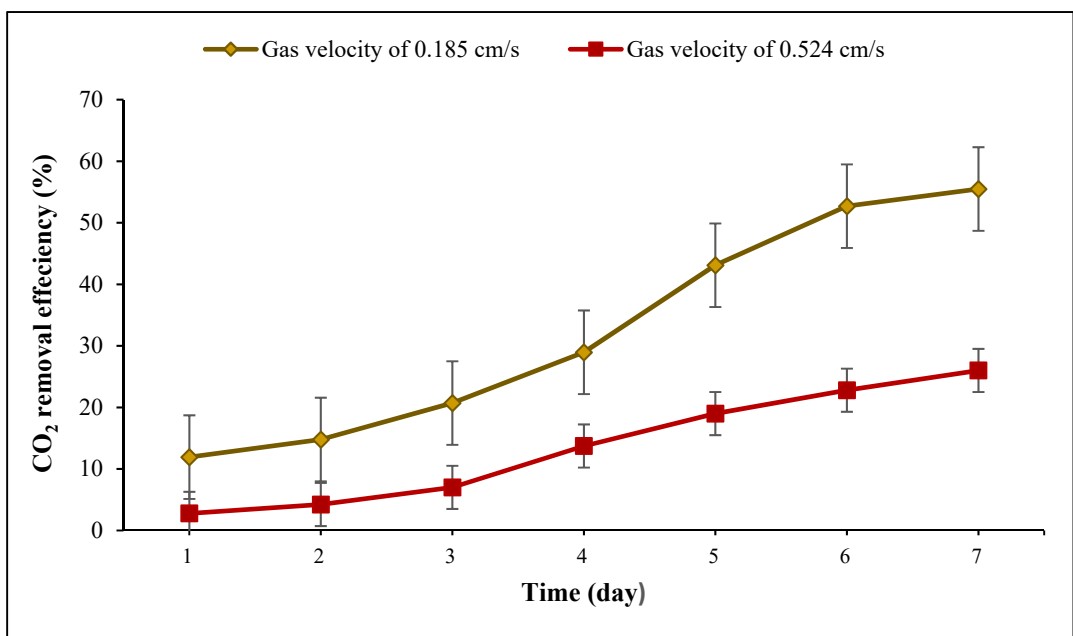

**Figure 8.** $CO_2$ removal efficiencies at gas velocities of (0.185 and 0.524 cm/s) during the growth of *Spirulina* sp. (temperature 30 ± 1 °C; pH = 9.5 ± 0.02).

**Table 5.** Summary of present study's results with other works.

| Gas Velocity | CO$_2$ Concentration (%) | Species | Maximum CO$_2$ Removal Efficiency (%) | Maximum Dry Weight (g L$^{-1}$) | Maximum Specific Growth Rate (day$^{-1}$) | Maximum Biomass Productivity (g L$^{-1}$ day$^{-1}$) | Reference |
|---|---|---|---|---|---|---|---|
| 0.185 cm/s [1]<br>0.524 cm/s | 5% | *Spirulina* sp. | 55.5%<br>23% | 0.86<br>1.62 (after 7 days) | 0.63<br>0.50 | 0.21<br>0.33 | This study |
| 8.0 L/min<br>12.0 L/min | 5% | *Chlorella protothecoides* | -<br>- | 0.32 (after 500 h)<br>0.24 (after 300 h) | -<br>- | -<br>- | [30] |
| 0.145 vvm<br>0.29 vvm | 0.038% | *Chlorella vulgaris* | 28%<br>2% | -<br>1.42 (after 14 days) | -<br>0.16 | -<br>- | [56] |
| 50 mL/min<br>70 mL/min | 1.75% | *Chlorella* sp. | 95.45%<br>88.63% | 2.5 (after 12 days)<br>2.8 (after 13 days) | 1.11<br>1.00 | -<br>0.17 | [57] |
| 0.74 cm/s<br>1.32 cm/s | 2% | *C. vulgaris* | 80%<br>64% | -<br>- | 0.24<br>0.18 (after 11 days) | -<br>- | [55] |
| 0.10 cm/s<br>0.50 cm/s | 4% | *C. vulgaris* | 14.6%<br>3.8% | 2.7<br>3.6 (after 10 days) | -<br>- | 0.41<br>0.47 | [60] |
| 0.1 vvm<br>0.5 vvm | 12% | *Scenedesmus obliquus* WUST4 | 67%<br>21% | -<br>- | -<br>- | -<br>- | [61] |

[1] To compare the gas velocity with the flow rate or vvm, the data was collected in Table 4.

*3.6. Perspective*

Due to the increase in CO$_2$ emissions and demand for better climate policies, this airlift reactor can be used in large scale. This is a cost-effective, sustainable way to capture carbon dioxide. Moreover, the results of this research illustrated that this airlift reactor was suitable for production of *Spirulina* biomass which is used in wide range.

## 4. Conclusions

This study demonstrated that the presence of cyanobacteria influenced the bubble velocity and gas holdup compared to the air-water system. It was observed that with an increase in the inlet gas velocity in the cyanobacterial culture, holdup and bubble velocity increased but more slowly than in the air-water system. Also, the inlet gas velocity and the average bubble size had a major impact on cyanobacterial growth and CO$_2$ uptake. At higher gas velocity, there was a decrease in cyanobacterial growth due to the interaction between bubbles and cells; as a result, the optimal gas velocity for *Spirulina* sp. growth was achieved at 0.524 cm/s. However, the maximum carbon dioxide efficiency (55.5%) was observed at the lower gas velocity (0.185 cm/s) due to the higher surface area of bubbles and longer residence time in the lower viscosity of the culture. Although CO$_2$ removal efficiency needs more investigation regarding different hydrodynamic parameters in the airlift photobioreactor, this study showed the feasibility of the CO$_2$ biofixation process using *Spirulina* sp.

**Author Contributions:** Data curation, A.P.T.; Investigation, Z.Z. and P.M.; Writing—review & editing, M.H.M. All authors have read and agreed to the published version of the manuscript.

**Funding:** This research received no external funding.

**Institutional Review Board Statement:** Not applicable.

**Informed Consent Statement:** Not applicable.

**Data Availability Statement:** All generated or analyzed data, the exploited softwares, and materials were included in this published article. The datasets generated and/or analyzed during the current study are available from the corresponding author on reasonable request.

**Conflicts of Interest:** The authors declare no conflict of interest.

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
