# Peer review of "Investigation of Hydrodynamic Parameters in an Airlift Photobioreactor on CO2 Biofixation by Spirulina sp."

_sustainability, doi:10.3390/su14127503_

Round 1

Reviewer 1 Report

In this article, the effect of hydrodynamic parameters on CO2 biofixation by Spirulina sp. in an airlift photobioreactor was investigated.

Specific comments and suggestions are listed below:

Spirulina is a genus of cyanobacteria…not algae, update the article.

Spirulina sp…where is the strain abbreviation?

Why was there such a big variation in temperature (30 ± 2) and pH (9.5 ± 2)? Such large differences will affect the growth of the culture.

Chlorophyll a has strong absorption features near 660-680 nm…why did you choose 680 nm for the cell density measurement?

It is better to measure dry weight every day, but not to use calibration lines.

Why only two different superficial gas velocities during the cyanobacterial growth (Figures 5 and 6) were selected?

The initial concentration of microalgae was about 0.5 g/L; however, this is not visible from Figure 5.

In addition, the standard deviations at all points of each line of each figure is the same, which suggests that three experiments were not carried out.

How about nutrients removal during the growth of Spirulina?

Please provide specific growth rate and biomass productivity values.

Please use ANOVA to compare differences.

Author Response

Reviewer #1: In this article, the effect of hydrodynamic parameters on CO2 biofixation by Spirulina sp. in an airlift photobioreactor was investigated.

Specific comments and suggestions are listed below:

  • Spirulina is a genus of cyanobacteria…not algae, update the article.

  • Thanks for your helpful comments and insights. Cyanobacteria was written instead of microalgae.

-       Spirulina sp…where is the strain abbreviation?

  • The exploited Spirulina strain was obtained from a culture collection. There was not additional details about its strain. So, we have used the “Spirulina” in this manuscript.

-       Why was there such a big variation in temperature (30 ± 2) and pH (9.5 ± 2)? Such large differences will affect the growth of the culture.

  • I made a mistake when I was writing these errors. The errors of the temperature and pH for this experiment were ± 1°C and ± 0.02, respectively. Please see section 2.4.

The temperature usually varied ±1°C as a PID controller was used in the water bath.

-     Chlorophyll a has strong absorption features near 660-680 nm…why did you choose 680 nm for the cell density measurement?

  • The employed references for this optical density were added. According to Cadondon et al. [40] peak chlorophyll absorption occurs at 680 nm which can also be used as an indication of Spirulina Please see section 2.9.

-       It is better to measure dry weight every day, but not to use calibration lines.

  • Thank you for your suggestion. The Calibration line was removed.

-       Why only two different superficial gas velocities during the cyanobacterial growth (Figures 5 and 6) were selected?

  • Because these velocities made the bubbly flow (homogeneous regime), and optimum growth of Spirulina was observed in this range however when the velocity increased above 0.875 cm/s due to more turbulent the cell growth decreased and it led to cells death. This information was added to section 3.4.

-       The initial concentration of microalgae was about 0.5 g/L; however, this is not visible from Figure 5.

  • Thank you for your attention. This figure was drawn again. Due to some settling in the reactor after the inoculation, the dry cell weight decreased on day 1 compared to day 0. Also some cells got stuck on the reactor walls.

-       In addition, the standard deviations at all points of each line of each figure is the same, which suggests that three experiments were not carried out.

  • Thank you again for your attention. This mistake was edited.

-      How about nutrients removal during the growth of Spirulina?

  • As this experiment was carried out for 7 days only, it was deemed sufficient to record the growth using dry cell weight and not using nutrient removal. The equipment for nutrient concentration analysis was not available in our laboratory.

-       Please provide specific growth rate and biomass productivity values.

  • This information was added and their graphs were drawn. Please see Table 4 and Figures 6 and 7.

-       Please use ANOVA to compare differences.

  • The requested information was added. Please see section 3.4 and 3.5.

Reviewer 2 Report

This work is informative. Manuscript preparation work should be improved.

  1. The paper title is “effect of…”, however, no effect results are shown in the section Abstract.
  2. Give your obtained values of dynamic parameters in section Abstract.
  3. Page 2: “it is affected by the superficial gas velocity, also known as gas flow rate.” What is “it”?
  4. Page 2: “is a lack of data and studies on the effect of these parameters and their combination”. What are “these”?
  5. Page 3: Why superficial gas velocities of 0.185 to 1.936 cm/s were tested?
  6. Explain the functions of “Gas holdup” in section 1?
  7. Section 2: Give a table to summarize all the hydrodynamic parameters tested.
  8. Figure 1: Strange terms “riser” and “downcomer”. What are their functions? How could Fig. 4 have their values?
  9. P4-5: Why “size also increased, but the mean bubble diameter decreased”?
  10. Section 3.4: Why only two superficial gas velocities (0.185, 0.524 cm/s) were tested? Because only these two velocities were tested, you could not conclude that “a maximum dry cell weight of 1.62 g/L at the superficial gas velocity of 0.524 cm/s”.
  11. Figures 3-6: What do you mean for “the number of experiments”?
  12. Give a table to compare your results with other works.
  13. Give a section to show how to apply your experimental results in fields.

Author Response

Reviewer #2: This work is informative. Manuscript preparation work should be improved.

  1. The paper title is “effect of…”, however, no effect results are shown in the section Abstract.
  • Thank you for your attention. This abstract was revised to include more effects, including numerical results.

  1. Give your obtained values of dynamic parameters in section Abstract.
  • The request information was added.

  1. Page 2: “it is affected by the superficial gas velocity, also known as gas flow rate.” What is “it”?
  • It is gas holdup, which was edited in the text. Please see section 1.

  1. Page 2: “is a lack of data and studies on the effect of these parameters and their combination”. What are “these”?
  • These are hydrodynamic parameters (gas holdup, bubble diameter, gas and liquid velocities) which were added. Please see section 1.

  1. Page 3: Why superficial gas velocities of 0.185 to 1.936 cm/s were tested?
  • To find out the bubbly flow range (homogenous regime) which was important for cyanobacterial growth. This information was added in section 2.3.

  1. Explain the functions of “Gas holdup” in section 1?
  • Thank you again. The functions of gas holdup were described in section 1.

  1. Section 2: Give a table to summarize all the hydrodynamic parameters tested.
  • Thank you for your suggestion. Table 2 was provided for this purpose.

  1. Figure 1: Strange terms “riser” and “downcomer”. What are their functions? How could Fig. 4 have their values?
  • Riser and downcomer are two interconnected zone in airlift reactors which riser is the internal loop and the most important part of the reactor where the most mixing and mass transfer occurs there. Downcomer is the external tube where the density in this part is higher than the riser, which causes the fluid to move. This information was added in section 1. Also Figure 1a was replaced with a new one to illustrate these elements better.
  • Thank you for your attention. To calculate liquid velocity in the riser, Equation 5 was used and then the liquid velocity in the downcomer was determined with Equation 6. (Ariser = 78.54 cm2 and Adowncomer = 98.17 cm2). Besides, the importance of the liquid velocity in these regions was explained in section 1.

  1. P4-5: Why “size also increased, but the mean bubble diameter decreased”?
  • As gas velocity increased, bubble size also increased (larger bubbles were released from the sparger at the bottom), but as these large bubbles rose in the liquid column, they quickly broke down into small bubbles, as a result the number of smaller bubbles increased and the mean bubble diameter decreased. This explanation was added in section 3.1.

  1. Section 3.4: Why only two superficial gas velocities (0.185, 0.524 cm/s) were tested? Because only these two velocities were tested, you could not conclude that “a maximum dry cell weight of 1.62 g/L at the superficial gas velocity of 0.524 cm/s”.
  • Because these velocities made the bubbly flow (homogeneous regime), and optimum growth of Spirulina was observed in this range, however when the velocity increased above 0.875 cm/s due to more turbulent the cell growth decreased and it led to cells death. This information was added to section 3.4.

  1. Figures 3-6: What do you mean for “the number of experiments”?
  • Thanks again for your comment. This mistake was edited. I wanted to say how many times this experiment was carried out.

  1. Give a table to compare your results with other works.
  • Table 4 was prepared to compare our results with other works.

  1. Give a section to show how to apply your experimental results in fields.
  • Thank you for your brilliant suggestion. This idea was added in the result and discussion section.

Round 2

Reviewer 1 Report

Please update:

1) Table 2 - please fill.

2) Figure 6 - All specific growth rates are calculated for an uninhibited growth during exponential phase (one value, not in dynamic)

3) Figure 7 - All productivity values are calculated for the duration of the experiment (i.e., from inoculation until final harvest) (one value, not in dynamic).

or remove both these figures and update table 4.

Author Response

(The authors gave the same response as above.)

Reviewer 2 Report

1. Paper title: Change “Airlift” into “airlift”.

2. Please consider the figure significance for the values of dynamic parameters.

Author Response

(The authors gave the same response as above.)
